# Bio-Based Hotmelt Adhesives with Well-Adhesion in Water

**DOI:** 10.3390/polym13040666

**Published:** 2021-02-23

**Authors:** Xi Yu, Chuang Dong, Wei Zhuang, Dongjian Shi, Weifu Dong, Mingqing Chen, Daisaku Kaneko

**Affiliations:** Key Laboratory of Synthetic and Biological Colloids, Ministry of Education, School of Chemical and Material Engineering, Jiangnan University, 1800 Lihu Avenue, Wuxi 214122, China; 6180608023@stu.jiangnan.edu.cn (X.Y.); 6180608009@stu.jiangnan.edu.cn (C.D.); 6190609025@stu.jiangnan.edu.cn (W.Z.); djshi@jiangnan.edu.cn (D.S.); wfdong@jiangnan.edu.cn (W.D.)

**Keywords:** bio-based, wet adhesion, bio-inspired, hot melt adhesive

## Abstract

We suggest a simple idea of bio-based adhesives with strong adhesion even under water. The adhesives simply prepared via polycondensation of 3,4-dihydroxyhydrocinnamic acid (DHHCA) and lactic acid (LA) in one pot polymerization. Poly(DHHCA-*co*-LA) has a hyperbranched structure and demonstrated strong dry and wet adhesion strength on diverse material surfaces. We found that their adhesion strength depended on the concentration of DHHCA. Poly(DHHCA-*co*-LA) with the lowest concentration of DHHCA showed the highest adhesion strength in water with a value of 2.7 MPa between glasses, while with the highest concentration of DHHCA it exhibited the highest dry adhesion strength with a value of 3.5 MPa, which was comparable to commercial instant super glue. Compared to underwater glues reported previously, our adhesives were able to spread rapidly under water with a low viscosity and worked strongly. Poly(DHHCA-*co*-LA) also showed long-term stability and kept wet adhesion strength of 2.2 MPa after steeping in water for 1 month at room temperature (initial strength was 2.4 MPa). In this paper, Poly(DHHCA-*co*-LA) with strong dry and wet adhesion properties and long-term stability was demonstrated for various kinds of applications, especially for wet conditions.

## 1. Introduction

Commercial glues are normally designed for use in dry conditions and do not have high water resistance. This feature causes corrosion of adhesives. For adhesion science and technology, it is a great challenge to investigate adhesives with high adhesion force and stability in a wet environment [1,2,3,4,5]. There are many attempts to achieve high performance under the conditions, however, few adhesives are able to act toughly because of difficulties of removing the water molecules from the interfaces, which prevents close contact between adhesives and substrates surfaces [6,7,8]. Insufficient contact produces a gap where a water molecule stays and reduces surface energy of the substrate which is the most important driving force for tough bonding [9,10,11,12,13]. The host-guest strategy was reported to prepare underwater adhesive after modifying the surface of substrates in advance [14], however, most of these adhesives cannot bind strongly without preprocessing the substrates [15,16,17,18].

It is found that mussels, barnacles, sandcastle worms and many other species are able to secrete biological adhesives to bond onto various substrates in a wet environment, among which mussels are the most famous object for studying [19,20,21,22,23,24,25]. Mussels have attracted wonderful attention due to their special adhesion onto many kinds of material surfaces [26,27,28]. The glue proteins secreted by mussels harden rapidly into solid and transforms into water-resistant adhesives. These glues are formed by different mussel foot proteins (Mfp), among which several kinds of low molecule weight of foot proteins (Mfp-3F, Mfp-3S and Mfp-5) were found at the substrate interfaces [29,30,31,32]. Lots of studies for the proteins have revealed the presence of an extraordinary amino acid 3,4-dihydroxy-L-phenylalanine (DOPA), which is formed by modification of tyrosine called mussel adhesive proteins [33,34,35]. Mussels’ adhesive proteins can stick strongly onto the substrate surfaces under a variety of conditions such as dry, wet and marine environments, even for nonstick materials such as Teflon. DOPA has a special chemical structure called catechol group in its structure. Working principles of the catechol group in DOPA have been proposed as hydrogen bonding interactions on polar and metal oxide surfaces [7,12], coordination bond interactions on metal oxides and metallic surfaces [8,26] and hydrophobic interactions on nonpolar surfaces [36,37,38]. However, preparation of the adhesive proteins costs a lot, and a huge amount of mussel specimens are required for making commercial glue applications [39,40,41]. Thus, there is great demand to explore mussel-inspired adhesive polymers which can work in water as a mussel protein.

There has been a lot of research concerning bio-inspired adhesives which showed good adhesion properties in wet environment [42,43,44,45,46]. For example, inspired by DNA and RNA, hydrogels adhesive tackified by nucleobase were successfully explored [45,46] due to the complementary base pairing interaction, such as adenine–thymine, guanine–cytosine and adenine–uracil. The nucleobase hydrogels exhibited good adhesive behaviors on various solid materials, including biological tissues [6].

To achieve strong adhesion in aqueous and blood environments, a hyperbranched polymer with a hydrophobic main chain and catechol groups at their chain end were designed and synthesized by Michael addition reaction. The hyperbranched polymer demonstrated strong adhesion onto diverse substrates under various kinds of environments. Their hydrophobic parts self-aggregated to form coacervates rapidly and removed water molecules out of the interface to increase exposure of catechol groups onto the substrate surfaces [7]. In other research, a reversible underwater glue based on responsive dynamic covalent bonding was prepared with similar ideal design. Inside of the glue, the dynamic forming and breaking of cross-linked polymer networks were repeated by the reversible anthracene dimerization under photo or thermal stimuli, which led to reversible adhesion on substrates [47,48]. 

Compared to petroleum-based adhesives, bio-based adhesives have recently shown considerable potential for various applications due to their many advantages such as environmental friendliness, biodegradability and renewability [49,50,51]. In particular, non-formaldehyde bio-based adhesives are highly demanded and needed to be improved to enhance bonding strength, especially for wet conditions. Modifying a bio-based polymer is a common approach to enhance adhesion strength, in which grafting catechol into a bio-based macromolecule is the most straightforward way. The adhesion properties of catechol-based adhesives depend on the graft ratio of catechol, however, the space steric effect limits the effective modification [52,53,54].

In our previous study, a novel hyperbranched bio-polyester was prepared via thermal polycondensation with a catechol-based unit 3,4-dihydroxyhydrocinnamic acid (DHHCA) and a functional unit 3-(4-hydroxyphenyl) propanoic acid (4HPPA). This adhesive showed excellent adhesion properties with high Tg and Tm, but it did not work well under water conditions [55,56,57]. Recently, it was reported that a hydrophobic microenvironment formed by long aliphatic chains improved wet adhesion properties of catechol-based glues by preventing H_2_O molecule contact from catechol [58]. As a familiar bio-based hydroxy carboxylic acid, lactic acid (LA) is an AB type of functional monomer with one –COOH group (A) and one –OH group (B), which is a similar structure to 4HPPA. LA always applied to bio-based polymers with long and soft aliphatic chains, which could improve water resistance and decrease melting temperature of our adhesives.

In this experiment, LA as a functional monomer and DHHCA as a main adhesive monomer are chosen and copolymerized for the novel bio-based hot-melt type adhesives with good adhesion under water.

## 2. Experiment Section

### 2.1. Materials

3,4-dihydroxyhydrocinnamic acid (DHHCA), which can be derived from cinnamon, lactic acid (LA) and disodium hydrogen phosphate (NaH_2_PO_4_) as a catalyst were purchased form Aladdin Biochemical Technology Co., Ltd. (Shanghai, China). Acetone, acetic anhydride (Ac_2_O) and ethanol were purchased form Chinese Medicine Group Chemical Reagent Co., Ltd. (Shanghai, China). All chemicals were of analytical purity and used without further purification.

### 2.2. Synthesis of Poly(DHHCA-co-LA)

Poly(DHHCA-*co*-LA) polyester was synthesized by two step reactions in one pot by the following procedure, as shown in Scheme 1: The mixture of DHHCA and LA were heated at 90 °C and mechanically agitated for 12 h in the presence of acetic anhydride and a catalyst of sodium dihydrogen phosphate. Then, the mixture was heated to 150 °C for 2 h and further heated to 200 °C for 10 h under a pressure below 100 Pa to improve degree of polymerization, and we obtained Poly(DHHCA-*co*-LA). To obtain Poly(DHHCA-*co*-LA)-Ac, acetic anhydride (Ac_2_O) was further added to poly(DHHCA-*co*-LA) for replacing the –OH group of catechol to Ac group. All polymers dissolved in acetone and precipitated in ethanol. The precipitates were filtered and washed with ethanol, and then dried at 80 °C for 12 h. All polymers were solid state and had yellow-brown color at room temperature.

### 2.3. Characterization

The molecular weights of poly(DHHCA-*co*-LA)s were determined by gel permeation chromatography (GPC; Waters Co., USA) calibrated with polystyrene standards (eluent: THF). The NMR spectra were obtained by an AVANCE III HD spectrometer (BRUGG Co., Switzerland) operating at 400 MHz for ^1^H NMR and with deuterated solvent of dimethyl-d6 sulfoxide used. FT-IR spectra were recorded on a FT-IR spectrometer (Nicolet iS50 FT-IR, Thermo Fisher Co., USA). The thermal properties were determined by differential scanning calorimetry (DSC 204 F1, NETZHCS Co., Germany). Samples were scanned from −20 to 120 °C and the rate was 10 °C/min under nitrogen atmosphere with 20 mL/min. Glass transition temperatures (T_g_) were determined using generated cycle data. The shear adhesion test was carried out by a tensile testing machine (WDW 5A, Jinan Xinguang Testing machine manufacturing Company Limited).

### 2.4. Lap Shear Test

Tensile strength measurements of the adhesives were performed using a universal test machine. Glass, aluminum and steel substrates were chosen for test substrates after washing with acetone and ethanol carefully. The size of plates was regulated as 100 mm in length, 20 mm in width and 5 mm in thickness. They were sandwiched by the hot-melt method at 95 °C under preload of 15 kPa. The bonding area for each substrate was constant at 2.0 × 2.0 cm. The bonded samples were then cooled down to room temperature in water. A tensile test machine with a 5000 N load cell was used, and the tensile rate was 2 mm/min. Each sample was tested five times and got the average value after removing maximum and minimum values.

## 3. Discussion

For the desirable bio-based adhesive, adhesion strength is a vital factor to be considered. As mentioned above, the hyperbranched bio-based polyester adhesive containing 3,4-dihydroxyhydrocinnamic acid (DHHCA) with a catechol group showed strong adhesion because DHHCA had similar structure to DOPA [55,56].

As reported before, a polymeric structure composed of DHHCA (AB_2_ type monomer) and a functional monomer (AB type monomer) forms a hyperbranched architecture. DHHCA is a molecule that has strong interaction with various kinds of surfaces in our polyester adhesives. AB type of functional monomer, which is another molecule for copolymerization, can decide various kinds of physical properties of polyester adhesives such as glass-transition temperature (T_g_), melting temperature (T_m_), water resistance, adhesive strength (molecular weight, intermolecular entanglement), etc. Especially T_g_ and T_m_ are the most important factors for hot-melt type adhesives. The hot-melt type adhesive is used by following two procedures. (1) Heat the adhesive to higher than melting temperature and spread between substrates; (2) keep the substrates close until cooling down below melting temperature.

A copolymer of DHHCA and 3-(3-hydroxyphenyl) propionic acid (3HPPA) was previously reported by our group and had excellent adhesion force over 20 MPa and high biocompatibility, however, high melting temperature (100 °C) limited the wide application for our living body. Therefore, we have chosen lactic acid (LA) as the functional monomer, which gives a low melting temperature to adhesives due to the soft main chain.

As shown in Table 1, to explore the best preparation condition, a series of poly(DHHCA-*co*-LA)s were examined. To improve the degree of polymerization, copolymers were prepared with high temperature and vacuum. It was confirmed that T_g_ decreased from 40.6 °C to 37.3 °C with decreasing concentration of C_DHHCA_, and there was no obvious influence to T_g_ by addition of anhydride. As a result of shorter polymerization time and lower polymerization temperature, lower T_g_ was caused by decreasing molecule weight. 

As shown in Figure 1B,C, poly(DHHCA-*co*-LA) exhibited good solubility for acetone and was insoluble for water. With increasing concentration of H^+^, poly(DHHCA-co-LA) gradually lost their solubility and precipitated in 1M HCl solution, as shown in Figure 1D. When NaOH solution was added to the solution, the color of the solution became dark, as shown in Figure 1E, which meant that catechol was ionized.

The structures of poly(DHHCA-*co*-LA) were characterized by ^1^H NMR, and the obtained spectrum are shown in Figure 2, which suggested formation of the target polyester structure. A series of peaks marked as a, a’, a’’, in a chemical shift range of δ = 6.36–7.35 ppm were assigned to individual aromatic protons on DHHCA, as shown in the chemical structure on the same figure. Broad peaks marked as b, b’ at δ = 2.51–2.91 ppm were assigned to the methylene group protons in the aliphatic chain. The peak marked as c at δ = 4.87–5.77 ppm was assigned to the methylidyne group protons, d at δ = 0.97–1.68 ppm was assigned to the methyl group protons, e at δ = 9.51–9.86 ppm was assigned to the carboxyl group protons, f at δ = 8.59–8.87 ppm was assigned to the phenolic hydroxyl group protons, g at δ = 1.91–2.02 ppm was assigned to acetyl group protons at the end of the polyester structure and h at δ = 4.32–4.41 ppm corresponded to the alcoholic hydroxyl groups, respectively. The integral strength ratio of the four methylene protons to the three aromatic protons and the one methylidyne proton to three methyl protons well agreed with the proton ratio, indicating that poly(DHHCA-*co*-LA) was successfully prepared.

We have prepared two types of bonded samples of poly(DHHCA-co-LA) for measur-ing the shear adhesion force by a tensile test machine, as shown in Figure 3. At the same time, using adhesive in water was shown in Appendix A. Sample A was made under water conditions with 15 kPa preloading. When the water temperature increased to 95 °C, the adhesive melted and spread between the substrates under water. After cooling down to 25 °C, the adhesive became solid, and sample A was obtained. Sample B was prepared in a vacuum oven at similar temperatures. Figure 3C denotes a schematic illus-tration of the shear adhesion test.

Figure 4A shows the wet and dry adhesive strength of poly(DHHCA-*co*-LA) with different C_DHHCA_ between glass substrates. Poly(DHHCA-*co*-LA)-Ac showed the lowest value due to absence of the catechol group at their chain ends. It was confirmed that dry adhesion strength decreased from 3.5 MPa to 3 MPa with decreasing concentration of C_DHHCA_. On the other hand, with increasing concentration of LA, wet adhesion strength increased. The best performance of wet adhesion was 2.7 MPa with concentration of LA 98.8%. These results indicate that DHHCA has a role of adhesion and LA prevents ingress of water molecules due to its hydrophobicity.

We have also carried out measurements of wet and dry adhesion strength on steel (Fe) and aluminum (Al) substrates in the same manner as shown in Figure 4B. Both wet and dry adhesion force between substrates did not reach 1 MPa due to low affinity between OH, COOH group of poly(DHHCA-*co*-LA) with hydrophobic substrates. However, wet adhesion force stayed over 0.8 MPa, which could be used where extremely high stress was not required in a wet environment.

Here, we considered why poly(DHHCA-*co*-LA) showed good adhesion properties under water conditions. Firstly, hyperbranched poly(DHHCA-*co*-LA) contained a lot of catechol group at the chain ends, which created strong interaction by hydrogen bonding between the polymer and different OH/O groups of substrate surfaces. Secondly, the presence of hydrophilic (OH/COOH) groups in the polymer structure supported spreading of adhesive between the substrate interfaces under water. Thirdly, hydrophobic aliphatic main chain of poly(DHHCA-*co*-LA) prevented catechol contact from water molecules by hydrophobic micro-environment [48].

Figure 5 shows the water contact angle on poly(DHHCA-*co*-LA) spin-coated on a glass substrate. It was confirmed that a high concentration of LA increased hydrophobicity of the adhesives due to a hydrophobic microenvironment.

Preloading force and viscosity of melt-state adhesives have important factors for the adhesion process, which affect spreading speed of adhesives between substrates, therefore, the effect on pre-loading and viscosity for adhesion strength was considered. Figure 6A shows wet adhesion strength depending on preloading pressure. It was found that high pressure produced rapid spreading of melt-state adhesive between substrates and the substrates were well glued for short exposing time to water. Figure 6B shows the effect of melting temperature to the adhesion force at constant preloading pressure of 15 kPa. With increased ambient temperature, adhesion force increased due to low viscosity proportional to the temperatures. This was because low viscosity also affected rapid spreading of glue between substrates.

Another fact revealed in this experiment was that poly(DHHCA-*co*-LA) kept 0.5 MPa of adhesion force under 50 °C water, as shown in Figure 6B, which was within the range of temperature for use in a living body, suggesting it could be used where extreme stress was not required.

Because the working of adhesive is also affected by the ambient temperature, we have measured the strength under different temperature, as shown in Figure 7A. The maximum value of wet adhesion strength was 4.7 MPa at 0 °C. Adhesion strength linearly decreased with increasing ambient temperature, which related to thermodynamics behavior of polymers.

Common adhesives lose adhesion strength quickly when they are left in a wet environment due to hydrolyzing and swelling. Long-term stability in a wet environment of poly(DHHCA-*co*-LA)3 have been measured at room temperature, and the result is shown in Figure 7B. Poly(DHHCA-*co*-LA)3 kept over 80% strength ratio even after steeping in water for one month, and the value was 2.2 MPa. This long-term stability indicates that hydrophobic soft long chain in the main chain prevents water intrusion into adhesive body for a long time.

## 4. Conclusions

We prepared a novel hot-melt type of bio-based adhesives with hyperbranched structure, poly(DHHCA-*co*-LA) from plant-based monomers of DHHCA and LA. The highest dry adhesion strength was 3.5 MPa with the highest concentration of DHHCA. The best wet adhesion strength was 2.7 MPa with the highest concentration of LA. These results indicate that DHHCA had an adhesion role, while LA had a role of preventing the invasion of water molecules into the adhesive body. The adhesive also showed long-term stability in a water environment and kept 80% strength ratio (2.2 MPa) under steeping water after one month. It could be said that it was very valuable to achieve an adhesive force of more than a few megapascals under both dry and wet conditions with one adhesive without surface pretreatment of substrates.

## Data Availability

Not applicable.

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
