# Peer review of "Bio-Based Hotmelt Adhesives with Well-Adhesion in Water"

_polymers, 2021, doi:10.3390/polym13040666_

Round 1

Reviewer 1 Report

The work of Yu is interesting and reports a more environmentally strong adhesive based on polycondensation. For the work is more a communication than a full length research article but I contains sufficient novelty. Below some comments that need to properly addressed:

“Many attempts to achieve high performance under the conditions, however, …” Please check grammar

Check overall spelling please. (e.g. studing, thet, Figure5…)

Ref 49. Good to mention the wavelength. Aspect of dynamic behavior see Chem. Eng. J. 2020, 402, 126259.

Bottom page 2: please add a scheme.

2.2. Please highlight for a general reader why the low pressure

For a general reader it needs to be clear why the structure is hyperbranched. A figure can help. The AB2 aspect is only clear later on.

Table 1: units are lacking.

Reviewer 2 Report

This work deals with a novel adhesive based on biopolymers prepared in pot polymerization. From my point of view, it is an interesting manuscript, and it does present some important aspects. Nevertheless, there are some major corrections that need be improved in order to approve this paper for its publication, namely:

  1. I suggest checking the whole article because there are many mistakes with both, the English language and the English grammar. My suggestion is that a native language English speaker needs to review the manuscript.
  2. In Table 1, on page 4, in the first column, it is not clear the nomenclature employed for each sample. I did not find what do the numbers mean. Could the authors please explain the meaning of these numbers?
  3. The obtained Mn and Mw are reported in Table 1. Nevertheless, I did not find any mention within the manuscript about the obtained values. Could the authors please add an explanation about these results? How do these results are related to the obtained properties of these adhesives?
  4. Finally, on page 8 in the conclusions section, it is stated that authors synthesized a novel-type of bio-based adhesives from “plant-derived” monomers. I did not understand this term. Could the authors be more specific?
